# Clinical Impact of RANK Signalling in Ovarian Cancer

**DOI:** 10.3390/cancers11060791

**Published:** 2019-06-08

**Authors:** Verena Wieser, Susanne Sprung, Irina Tsibulak, Johannes Haybaeck, Hubert Hackl, Heidelinde Fiegl, Christian Marth, Alain Gustave Zeimet

**Affiliations:** 1Department of Obstetrics and Gynaecology, Medical University of Innsbruck, 6020 Innsbruck, Austria; verena.wieser@i-med.ac.at (V.W.); irina.tsibulak@i-med.ac.at (I.T.); heidelinde.fiegl@i-med.ac.at (H.F.); Christian.Marth@i-med.ac.at (C.M.); 2Institute of Pathology, Medical University Innsbruck, 6020 Innsbruck, Austria; Susanne.Sprung@i-med.ac.at (S.S.); Johannes.Haybaeck@i-med.ac.at (J.H.); 3Biocenter, Division of Bioinformatics, Medical University Innsbruck, 6020 Innsbruck, Austria; Hubert.Hackl@i-med.ac.at

**Keywords:** RANKL, RANK, OPG, inflammation, ovarian cancer, denosumab

## Abstract

Ovarian cancer (OC) is a gynaecological malignancy with poor clinical outcome and limited treatment options. The receptor activator of nuclear factor-κB (RANK) pathway, activated by RANK ligand (RANKL), critically controls bone metabolism, tumourigenesis and tumour immune responses. Denosumab, a monocloncal RANKL antibody, exerts tumour-suppressive effects in mice and humans. Here, we investigated the relevance of RANK signalling in OC. *RANK*, *RANKL* and *OPG* expression in 192 epithelial OC tissues was compared to expression in 35 non-malignant control tissues and related to clinico-pathological characteristics. Findings were validated in a cohort of 563 OC patients from The Cancer Genome Atlas (TCGA). The expression of RANK, RANKL and OPG was studied in four OC cell lines and the impact of RANK ligation or blockade on OC cell proliferation was determined. RANK, RANKL and OPG were expressed in epithelial and stromal cells in OC. *RANKL* expression was elevated in OC tissue, particularly in *BRCA1/2* mutated tumours. High *RANKL* expression independently predicted reduced progression-free (PFS, *p* = 0.017) and overall survival (OS, *p* = 0.007), which could be validated in the TCGA cohort (PFS, *p* = 0.022; OS, *p* = 0.046, respectively). Expression of *RANK* and *OPG* in OC cells was induced by inflammatory cytokines IL-1β and TNFα. Neither recombinant RANK ligation nor denosumab treatment affected OC cell proliferation. Our study independently links *RANKL* expression with poor clinical outcome in two unrelated OC cohorts. These findings implicate RANK signalling in the immunopathogenesis of OC and warrant clinical trials with denosumab in OC.

## 1. Introduction

Ovarian cancer (OC) is one of the most common cancers amongst women in Europe and the leading cause of death in gynaecological cancers [1]. In recent years, multiple treatment modalities have emerged including surgical therapy, chemotherapy, anti-angiogenic agents, and PARP-inhibitors (PARPis). However, prognosis remains devastating and highlights the necessity for a better knowledge about disease pathogenesis and involved pathways [2].

The interplay of receptor activator of nuclear factor-κB (RANK), its ligand RANKL and Osteoprotegerin (OPG) are established regulators of bone metabolism. Specifically, RANKL expressed on osteoblasts activates RANK signalling and transcriptional activation of osteoclasts which promote osteoclastogenesis and bone resorption [3]. The endogenous OPG functions as a soluble decoy receptor preventing RANKL from activating its receptor RANK [4]. Moreover, the RANK/RANKL/OPG system also plays essential roles in immunity and tumourigenesis [5,6] and RANK/RANKL cancer cell expression correlated with metastasis and tumour progression in human breast cancer (BC) [7]. More specifically, previous studies demonstrated that RANKL mediates progesterone-driven mammary carcinogenesis [8,9,10] and RANK/RANKL signalling controlled *BRCA1* mutation-driven mammary tumours [11]. Sigl et al. demonstrated that the genetic and pharmacological inhibition of RANKL in mice abolished the occurrence of *Brca1* mutation-driven pre-neoplastic lesions [11]. Further, RANK/RANKL blockade impaired proliferation and expansion of mammary progenitors from human *BRCA1* mutation carriers indicating a significant role of RANK/RANKL signalling in inherited BC. Therefore, BRCA-P, an international randomized phase III study (NCT01864798) investigates denosumab for the prevention of BC in *BRCA1* mutation carriers. Notably, mutations in the *BRCA1* gene are not only associated with inherited BC but also with OC. Although BC and OC present as distinct clinical entities, recent evidence supported a substantial overlap with respect to genetic and epigenetic alterations [12].

Furthermore, the inhibition of RANK signalling was demonstrated to improve the effectiveness of checkpoint blockade in experimental solid tumours, experimental metastasis and malignant melanoma patients [13,14,15]. This effect seems to depend on a functional immune system as the depletion of natural killer cells and T cells abrogates this effect in mice [16]. In line with this, a window of opportunity trial including pre-menopausal women with early BC, RANK pathway inhibition using denosumab resulted in a significant increase in TILs (NCT01864798), however, had less effect on tumour proliferation.

Regarding the potential relevance of RANK/RANKL in hereditary BC and in the immunopathogenesis of experimental tumours and malignant melanoma patients, we hypothesized a differential regulation of this pathway in OC. We, therefore, determined *RANK*, *RANKL* and *OPG* expression in 192 OC tissues and 35 non-malignant control tissues and performed association analysis with clinicopathological characteristics and clinical outcome of OC patients. To specifically understand RANK signalling in *BRCA1/2* driven OC, tissue was separately analysed according to *BRCA1/2* mutation status determined in OC tissue. To further strengthen our hypothesis, results were validated in a larger TCGA Affymetrix cohort comprising 563 OC patients (ovarian cystadenocarcinomas). Using four OC cell lines, inhibition and, vice versa, activation of the RANK pathway was performed to investigate a direct effect of RANK signalling on tumour cell proliferation.

## 2. Results

### 2.1. RANKL Is Highly Expressed in OC and BRCA1/2 Mutated Tumours

To assess a role for the RANK/RANKL/OPG pathway in OC, we analysed the expression in 192 OC samples by qPCR. Transcript levels were compared to 21 non-malignant ovaries and 14 non-malignant fallopian tubes of healthy controls. We found highly elevated *RANKL* and *OPG* expression in OC tissue compared to control tissues (Figure 1A,B), which was largely unrelated to histological subtype (Appendix A). *RANK* was equally expressed in OC and non-malignant fallopian tubes, but lower expressed in non-malignant ovaries (Figure 1C). Spearman rank association analyses demonstrated a significant positive correlation between *RANK, RANKL* and *OPG* expression (*RANK* and *RANKL*: *rs* = 0.164, *p* = 0.018, Appendix A; *RANKL* and *OPG*: *rs* = 0.167, *p* = 0.016, Appendix A; *RANK* and *OPG*: *rs* = 0.214, *p* = 0.002, Appendix A).

In a next step, *RANK, RANKL* and *OPG* expression was stratified according to *BRCA1/2* mutation status available for 190 patients. In this cohort, 44 patients (22.9%) exhibited a *BRCA1* or *BRCA2* mutation. Notably, *BRCA1/2* mutated cancers exhibited a higher *RANKL* expression (*p* = 0.033, Figure 1D), but not increased *RANK* or *OPG* expression (Appendix A). More specifically, *RANKL* expression was significantly higher in *BRCA1* mutated tumours (*n* = 35) compared to *BRCA* wild-type (wt) tumours and also elevated in *BRCA2* mutated tumours (*n* = 9) which did not reach statistical significance (Appendix A). Furthermore, we performed Spearman rank analysis for transcriptional levels of *BRCA1/2* and *RANK/RANKL/OPG*. We found a direct correlation between *BRCA2* and *RANKL* expression (*rs* = 0.309, *p* < 0.001; Appendix A) and *BRCA2* and *OPG* expression (*rs* = 0.230, *p* = 0.002; Appendix A) but no correlation between *BRCA1* and *RANK/RANKL/OPG* levels.

### 2.2. High RANKL mRNA Expression is Associated with Poor Prognosis

Based on the observation that OC patients exhibited increased *RANKL* expression, we next assessed an impact of *RANK*, *RANKL* and *OPG* expression on the clinical outcome of OC patients. We utilized the Youden Index to dichotomize the OC cohort into “high” and “low” *RANK*, *RANKL* and *OPG* expressing tumours [17]. Univariate survival analysis (Table 1) demonstrated that patients with low *RANKL* expression (<62nd percentile) in OC tissue exhibited a median progression-free survival (PFS) of 3.6 years (95% confidence interval (CI): 1.8–5.3) whereas patients with high *RANKL* (>62nd percentile) expression exhibited a median PFS of only 1.7 years (95% CI: 1.2–2.2) (*p* = 0.010; Figure 2A). There was also a strong association between high *RANKL* expression in OC tissue and impaired overall survival (OS): Patients with low *RANKL* expression (<62nd percentile) exhibited a median OS of 8.8 years (95% CI: 6.0–11.6) while patients with high *RANKL* expression (>62nd percentile) exhibited a median OS of 3.6 years (95% CI: 2.5–4.7), (*p* = 0.005; Figure 2B). This prognostic impact was unrelated to *BRCA1/2* mutation status as we observed comparable results in the subgroup of patients with *BRCA1/2* wt tumours (Appendix A). *BRCA1/2* mutation status, however, was not associated with clinical outcome in our OC cohort. When analysing the subgroup of patients suffering from HGSOC, which represents the majority of OC patients in our cohort, we observed similar results (Appendix A).

Similar findings were notable for *RANK* expression in our cohort (Table 1): Patients with low *RANK* expression (<25th percentile) in OC tissue exhibited a median PFS of 3.6 years (CI: 0.9–6.2) whereas patients with high *RANK* expression (>25th percentile) exhibited a median PFS of only 2.0 years (CI: 1.5–2.4) (Figure 2C), however, this did not reach statistical significance (*p* = 0.117). In contrast, patients with low *RANK* expression (<25th percentile) in OC tissue exhibited a median OS of 9.3 years (CI: 2.0–16.6) while patients with high *RANK* expression (>25th percentile) exhibited a median OS of 3.71 years (CI: 1.85–5.57), (*p* = 0.030; Figure 2D). Importantly, multivariate analyses identified *RANKL,* but not *RANK,* as an independent prognostic factor for PFS (HR 1.42, *p* = 0.017; Table 1) and OS (HR 1.70, *p* = 0.007; Table 1). *OPG* expression (optimal cut-off: 92nd percentile) was not of prognostic significance regarding PFS or OS in our cohort (Table 1, Appendix A).

### 2.3. Validation of the Prognostic Impact of RANKL Expression in an Independent Cohort

Gene expression datasets from The Cancer Genome Atlas (TCGA) project on primary serous ovarian carcinomas (*n* = 563) were analysed for the prognostic impact of *RANKL* expression on PFS and OS. High *RANKL* expression was associated with decreased PFS (*p* = 0.040, Figure 3A) and OS (*p* = 0.036, Figure 3B). Notably, also in the TCGA cohort, multivariate analyses identified *RANKL* as an independent prognostic factor for PFS (HR 1.32, *p* = 0.022; Table 2) and OS (HR 1.26, *p* = 0.046; Table 2). We did not find an impact of *RANK* or *OPG* expression on OC prognosis in the TCGA cohort.

We also performed a reverse analysis: We calculated the Youden Index for *RANKL*, *RANK* and *OPG* expression in the TCGA set and then applied this cut-off to the clinical analysis of both cohorts. Indeed, using the cut-off from the TCGA cohort, *RANKL* expression was associated with worse PFS (*p* = 0.032; Appendix A) and OS (*p* = 0.026; Appendix A) in the TCGA cohort, and also in our OC cohort (*p* = 0.013 and *p* = 0.007, respectively; Appendix A). In contrast, *RANK* and *OPG* expression were not associated with worse OS or PFS in the TCGA cohort.

### 2.4. RANK, RANKL and OPG Are Expressed on Tumour Cells and in the Tumour Microenvironment

To elucidate the intratumour distribution of RANK, RANKL and OPG we randomly selected 20 OC tissues for immunohistochemistry. Epithelial cells and, to some extent, stromal cells expressed RANK, RANKL and OPG in healthy and diseased tissue (Figure 4, Appendix A). Notably, non-malignant fallopian tubes expressed RANK, RANKL and OPG, which was pronounced in OC tissue (Figure 4). We did not find an association for semi-quantitative immunohistochemical RANK, RANKL and OPG expression patterns and clinicopathological characteristics and also tumour grade or histological subtype were not associated with a specific distribution pattern of RANK, RANKL or OPG expression. Additional representative images of IHC are shown in Appendix A.

### 2.5. Neither Stimulation with RANKL Nor Treatment with Denosmuab Affects OC Cell Proliferation

Similar to OC tissue, OC cell lines SKOV6, HOC7, HTB77, and OVCAR3 exhibited *RANK, RANKL* and *OPG* expression at baseline (Appendix A). Expression of both, *RANK* and *OPG,* was induced in SKOV6 cells by inflammatory mediators such as IL-1β or TNFα but not mitogenic mediators such as IL-6 or LPA (Appendix A). In contrast, *RANKL* expression was largely unaffected by cytokine stimulation (Appendix A). Notably, progesterone did not impact on *RANKL* expression (Appendix A), as previously demonstrated for breast cancer cells [18].

Considering the known effect of denosumab on cell proliferation in vitro [18,19] we next assessed whether pharmacological RANKL inhibition or stimulation with recombinant RANKL could directly influence cell proliferation of OC cells. Therefore, we treated OC cell lines with increasing concentrations of denosumab with or without cisplatin co-treatment. Denosumab did not affect the proliferation of OC cell lines OVCAR3 (Appendix A), SKOV6 (Appendix A) or HTB77 (Appendix A) evaluated by MTT assays and by expression analyses of cell the cycle proteins G1/S-specific cyclin-E (*CCNE)* and transcription factor E2F3a (*E2F3A*). Furthermore, denosumab in combination with cisplatin did not have an additional effect on platinum-induced cytotoxicity in OVCAR3 (Appendix A), SKOV6 (Appendix A) and HTB77 cells (Appendix A). Despite expression of RANK in OC cells, stimulation with recombinant RANKL did not affect cell proliferation (Appendix A).

### 2.6. High CCND1 mRNA Expression is Associated with Poor Prognosis

To further investigate downstream signalling of RANK activation we performed correlation analysis between *RANK* and *RANKL* and mediators of either the NFκB pathway (*CCND1*) or the ID2 pathway (*P21*) [20] in the TCGA cohort. As we found a correlation between both *RANK* and *RANKL*, and *CCND1* (Figure 5A,B) we performed Kaplan—Meier survival analysis according to high (>64th percentile) and low *CCND1* (<64th percentile) expressions and demonstrate that high *CCND1* is associated with reduced PFS (*p* = 0.208; Figure 5C) and OS (*p* = 0.013; Figure 5D). In contrast, we did not find a correlation between *RANK/RANKL* and *P21* expression These data suggest that downstream RANKL mediator(s), specifically CCND1, contribute to poor clinical outcome in OC.

## 3. Discussion

In this study, we investigated the RANK/RANKL/OPG pathway in OC and its impact on clinical outcome in two unrelated OC cohorts. We found that *RANK, RANKL* and *OPG* were highly expressed in OC tissue which localized to tumour epithelium and to tumour stromal cells. Previous studies demonstrated that a high percentage of tumours such as breast, cervical, endometrial, colorectal, prostate cancer and malignant melanoma express RANK, and some of them also express RANKL [6,21,22,23].

As little is known about the prognostic impact of RANK and RANKL expression in OC, we analysed our cohort of 192 OC patients including various histologic types and validated our findings in an independent TCGA cohort comprising serous ovarian carcinomas. These studies demonstrate that high *RANKL* expression independently predicted poor clinical outcome defined by a shorter PFS and OS in a multivariate analysis. Considering different histologic subtypes included in our and the TCGA cohort, a subgroup analysis of HGSOC patients (*n* = 122) in our cohort revealed similar results (Appendix A) compared to the entire cohort including various histological subtypes. RANK and/or RANKL expression previously associated with poor clinical outcome in gastric cancer [24], acute myeloid leukaemia [25], breast cancer [26,27], (clear-cell) renal cell carcinoma [28,29], prostate cancer [30] and osteosarcoma [31].

RANK signalling is critically involved in *BRCA*-driven tumourigenesis in BC [11] and previous studies demonstrated that RANKL mediates proliferation of the mammary epithelium in mice [32] and humans [10]. Treatment of non-malignant human breast epithelial cells with denosumab significantly decreased the frequency of colony formation [19]. In line with previous findings in BC [11], we noted that *RANKL* expression was particularly elevated in *BRCA1/2* mutated OC as compared to *BRCA1/2* wild-type (wt) tumours. Future studies should investigate a functional role for RANK/RANKL/OPG signalling in DNA repair mechanisms to foster our biological understanding how RANK signalling impacts on the course of OC [33]. In this regard, we did not note that modulation of RANK signalling (by denosumab or recombinant RANK ligand) affect OC cell proliferation in four OC cell lines. As such, we speculate that OC-derived RANKL may modulate the tumour microenvironment and particularly tumour immune responses [7] which could involve stromal cells or infiltrating immune cells and an inflammatory OC environment [34,35,36].

In line with this, pro-inflammatory mediators such as IL-1β and TNFα induced *RANK*, and *OPG* expression in OC cell lines. As we observed a correlation between *CCND1* and both, *RANK* and *RANKL* expression in the TCGA cohort and we noted an association between *CCND1* and reduced OS, we speculate that NFkB activation may be a critical event downstream of RANK activation in OC. Moreover, RANKL acts as a chemoattractant to M2 macrophages and tumour cells [6,37] and the RANK pathway is involved in epithelial–mesenchymal transition (EMT) and stemness, and facilitates tumour growth and metastasis by modulating immune and vascular niches [38]. RANKL also up-regulates the angiogenic process by stimulating the proliferation and survival of endothelial cells [39,40] and may promote extravasation/intravasation of RANK-expressing cancer cells, as well as their migration to distant organs [38]. More recently, Khan et al. demonstrated that RANKL blockade can rescue melanoma-specific T cells from thymic deletion and, therefore, increases the anti-tumour immune response in melanoma [41]. These data suggest also a potential role of the RANK pathway in immune escape and immunotherapy which may become relevant in OC [42,43].

More precisely, immunotherapy for the treatment of OC is promising as previous studies demonstrated that the presence of tumour infiltrating lymphocytes (TILs) is associated with improved clinical outcome in OC patients [44,45]. Therapies targeting TILs in OC include immune checkpoint blockade, cancer vaccines, and adoptive cell therapy [43]. However, there are currently no approved immunotherapies for OC. Interestingly, short course of denosumab in the D-BEYOND study including premenopausal women with early BC significantly increased stromal TILs without reducing tumour proliferation rate (NCT01864798). These data are in line with our hypothesis that the RANK pathway may influence OC biology via immunomodulation. Notably, latest studies demonstrate that RANK signalling inhibition may improve the effectiveness of checkpoint blockade in cancer treatment [15]. Ahern et al. demonstrated that RANKL blockade improved the efficacy of anti-CTLA4 monoclonal antibodies against experimental solid tumours and experimental metastases [13]. Furthermore, Smyth et al. demonstrated that that a combination of natural killer cells and T cells are required for the antitumour activity of anti-CTLA-4 and anti-RANKL in mice [16]. A retrospective review by Afzal et al. demonstrated that the combination of immune checkpoint inhibitors and denosumab improved median PFS and OS malignant melanoma patients compared to checkpoint inhibitors alone [14]. In OC, checkpoint inhibitors have not been convincing as the observed response rates were low, however, improvements in therapeutic efficacy have been proposed through combination with PARPis and other immunostimulatory compounds. Considering that RANK inhibition was demonstrated to improve immune responses the combination of RANK inhibitors, e.g., denosumab, with checkpoint inhibitors may be worth to assess in clinical trials of OC and other gynaecological malignancies, such as cervical cancer, which was shown also to be of high expression of RANK and RANKL [46].

In BC, on the one hand RANK signalling (driven by progesterone) controls the onset of hormone-induced BC through the expansion of mammary progenitor cells and on the other hand, RANK and RANKL also critically regulate *BRCA1*-mutation-driven BC. The latter could be explained by the fact that common variations in the RANK gene modify the risk of developing BC in *BRCA1*-mutation carriers. Further, women carrying a germline *BRCA1* mutation show high levels of progesterone and oestrogen during the luteal linking *BRCA1* associated tumourigenesis to female sex hormones [20]. Considering that progesterone triggers massive induction of RANKL in mammary-gland epithelial cells, genetic inactivation of RANK prevents progestin-driven epithelial proliferation [10,32,47] and we detected progesterone receptors in our OC cohort (0–90% PR positive for IHC), we tested the “in vitro” impact of progesterone on *RANKL* expression in OC cell lines. However, progesterone treatment did not induce expression of RANKL/RANK/OPG. This is in contrast to findings in BC cells [10,32,47] and appears uncoupled from the observation that progesterone protects against OC [48].

OC is a clinically challenging cancer entity that requires intensive treatment combining surgery, chemotherapy and angiogenic inhibitors [49]. However, prognosis of OC patients remains devastating, which is why novel therapeutic options are eagerly awaited [2]. Denosumab is a neutralizing RANKL antibody that has been approved by the FDA for the treatment of osteoporosis. Moreover, denosumab is currently used to treat bone fractures of metastatic disease and RANKL inhibition has been extensively studied in preclinical tumour models [47] and is now proven as a preventive strategy for women carrying *BRCA1* mutations and high risk of BC [20].

## 4. Materials and Methods

### 4.1. Patients and Samples

Ovarian tissue samples from 192 patients with OC obtained at primary debulking (patients were 24 to 90 years old; median age at diagnosis was 59.7 years) and control tissues from 35 patients obtained by elective salpingectomy or elective oophorectomy for benign conditions (i.e., salpingectomy for sterilization or contralateral salpingo-oophorectomy as part of surgery for benign cysts) (patients were 30.4 to 74.3 years old, median age: 51.0 years) were collected and processed at the Department of Obstetrics and Gynaecology of the Medical University of Innsbruck, Austria between 1989 and 2010 as described recently [42]. Written informed consent was obtained from all patients before enrolment. The study was reviewed and approved by the Ethics committee of the Medical University of Innsbruck (reference number: 1157/2018) and conducted in accordance with the Declaration of Helsinki. All samples were anonymized before the commencement of the analysis. All patients were monitored within the outpatient follow-up program of our department. Tumour specimens were analysed for somatic *BRCA1/2* mutations as previously described [50]. The median observation period was 1.6 years (0.03–22 years) regarding PFS and 3.6 years (0.09–26.1 years) concerning the OS. All patients were of Caucasian ethnicity. Clinicopathological features are shown in Table 1.

### 4.2. RNA Isolation and Reverse Transcription

Total cellular RNA extraction from tissue samples and in vitro experiments and reverse transcription were performed as previously described [42]. Histologically, ~75% of the analysed tissue samples consisted of cancer cells (Appendix A) did not correlate with *RANK*, *RANKL* and *OPG* expression (Appendix A).

### 4.3. Quantitative Real Time PCR

Primers and probes for the TATA box-binding protein (*TBP*; endogenous RNA-control) were used according to Bieche et al. [51]. Primers and probes for *RANK, RANKL* and *OPG* were purchased from Applied Biosystems (Hs00921372_m1, Hs00243522_m1, Hs00900358_m1, Foster City, CA, USA). The following primer sequences were used for *CCNE*: for 5′ ACT TAA GGG CCT TCA TAA TCA TTA ATT C 3′, rev 5′ GCA GCC AAA CTT GAG GAA ATC TAT 3′, probe 5′ FAM-AGA ATT TCA TCT CCT GAA CAA GCT CCA TCT GTC-TAMRA 3′; and *E2F3A*: for 5′ TTT AAA CCA TCT GAG AGG TAC TGA TGA 3′, rev 5′ CGG CCC TCC GGC AA 3′, probe 5′ FAM-CGC TTT CTC CTA GCT CCA GCC TTC G_TAMRA 3′. PCR reactions were performed as previously described [42]. *BRCA1* and *BRCA2* mRNA expression was determined as previously described [50].

### 4.4. TCGA Cohort

Analyses were performed on The Cancer Genome Atlas (TCGA, National Cancer Institute, Bethesda, MD, USA) publicly available dataset. Eligible patients were those who were defined as having serous ovarian cystadenocarcinoma in the TCGA dataset and who had complete information on age at OC diagnosis, tumour grade, FIGO stage, survival, and had gene expression analyses available.

### 4.5. Immunohistochemistry

Immunohistochemistry was performed using an automated immunostainer (BenchMark ULTRA, Ventana Medical Systems, Tucson, AZ, USA). In short, formalin-fixed, paraffin-embedded (FFPE) tissue sections were prepared with cell conditioning reagent for antigen retrieval. Anti-SPATA2 antibody (Sigma-Aldrich, HPA048581, St. Louis, MO,) was incubated for 30 min at 37 °C and for visualization the Ultra View DAB Detection Kit (Ventana Medical Systems, Oro Valley, AZ, USA) was used as recommended. Slides were counterstained with haematoxylin and bluing reagent. Images were acquired with a Zeiss AxioCam (Oberkochen, Germany).

### 4.6. Culture and Treatment of OC Cells

OVCAR3, HOC7, SKOV6 and HTB77 human ovarian cancer cells were purchased from ATCC (Middlesex, UK) and cultured in RPMI supplemented with 10% foetal bovine serum and penicillin/streptomycin. We have performed the STR profiling in our lab in February 2018 before performing in vitro experiments and regularly performed Mycoplasma testing. Cells were treated with recombinant human IL-1β (10 ng/mL; Invivogen, San Diego, CA, USA), TNFα (25 ng/mL; Peprotech, Rocky Hill, NJ, USA), IL-6 (10 ng/mL; Peprotech, Rocky Hill, NJ, USA), LPA (20 μM; Sigma-Aldrich, St. Louis, MO, USA), progesterone (1 mg/mL; Proluton^®^, Sandoz, Holzkirchen, Germany), Denosumab (X-Geva^®^; Amgen, Thousand Oaks, CA, USA; 3 µg/mL, 30 µg/mL, 300 µg/mL), cisplatin (1–5 µg/mL, Sandoz, Holzkirchen, Germany), recombinant Human sRANK Ligand (*Escherichia coli*-derived; Peprotech, Rocky Hill, NJ, USA) for indicated time points.

### 4.7. MTT Assay

MTT assay was performed as previously described [52]. In short, MTT (10 mL/well, Sigma-Aldrich, St. Louis, MO, USA) was added to cell culture in 96 well plates after indicated timepoints and incubated for another 5 h. After the addition of 100 mL DMSO (Sigma-Aldrich, St. Louis, MO, USA), the extinction was measured photometrically (wavelength 550 nm).

### 4.8. Statistical Analysis

The non-parametric Mann–Whitney *U* test or Kruskal–Wallis test were applied to test for statistical significance between two groups or more than two groups, respectively. The correlations between *RANK, RANKL, OPG, BRCA1* and *BRCA2* mRNA expression were assessed by Spearman rank correlation analyses. PFS was defined as the time from diagnosis of the primary of tumour to the histopathological confirmation of recurrence or metastases or death from any cause and OS as the time from diagnosis of the primary of tumour to death from any cause. Patients were censored at the date of their last personal contact. Univariate Kaplan—Meier analyses and multivariable Cox survival analyses were used to explore the association of *RANK, RANKL* and *OPG* expression with PFS and OS (the *p*-value cut-off for inclusion to the multivariable Cox analysis was 0.2). For survival analyses, patients were dichotomized into low and high mRNA expression level groups by the optimal cut-off expression value calculated by the Youden Index [17]. Thereby, the cut-off was set at the 25th percentile for *RANK*, the 92nd percentile for *OPG*, and the 62nd percentile for *RANKL* which was further applied for the TCGA cohort. Experiments with more than two comparisons were tested for statistical significance by one-way ANOVA. *p*-values less than 0.05 were considered as statistically significant. Statistical analysis was performed using SPSS statistical software (version 20.0.0; SPSS Inc., Chicago, IL, USA).

## 5. Conclusions

RANKL expression is elevated in OC and independently predicts poor clinical outcome in two unrelated cohorts proposing a significant role of RANK signalling in the immunopathogenesis of OC. This clinical observation suggests that denosumab may be of therapeutic use in OC, although we acknowledge that further mechanistic studies about the role of RANK signalling in OC are warranted.

## Figures and Tables

**Figure 1 cancers-11-00791-f001:**
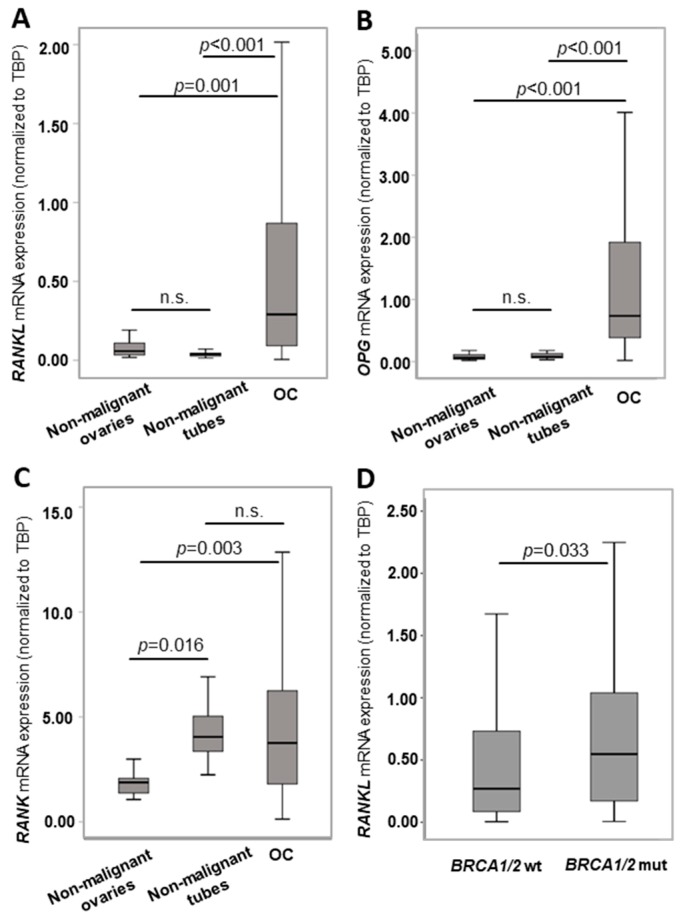
*RANK, RANKL* and *OPG* mRNA expressions are elevated in OC tissue, particularly in *BRCA1/2* mutated OC. (**A**) *RANKL*, (**B**) *OPG* and (**C**) *RANK* expression in non-malignant ovaries (*n* = 21), non-malignant fallopian tubes (*n* = 14) and OC (*n* = 192). (**D**) *RANKL* mRNA expression in *BRCA1/2* mutated (mut) OC (*n* = 44) compared to *BRCA1/2* wild-type (wt) tumours (*n* = 146). *RANK*, *RANKL* and *OPG* mRNA expression values were normalized to *TBP* expression, n.s., not significant.

**Figure 2 cancers-11-00791-f002:**
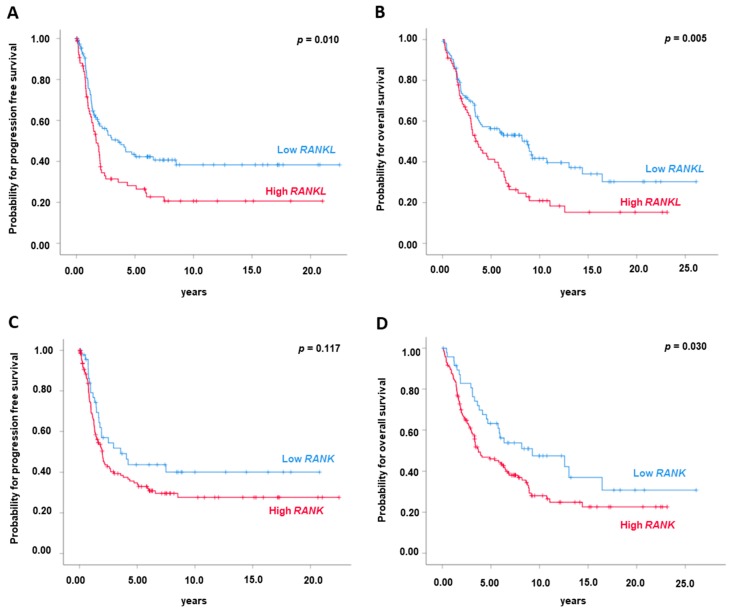
High *RANK* and *RANKL* mRNA expressions are associated with worse PFS and OS in OC. *RANKL* mRNA expression (*n* = 192) and (**A**) progression free survival and (**B**) overall survival. *RANK* mRNA expression (*n* = 192) and (**C**) progression free survival and (**D**) overall survival. *RANK* and *RANKL* mRNA expression values were normalized to *TBP* expression.

**Figure 3 cancers-11-00791-f003:**
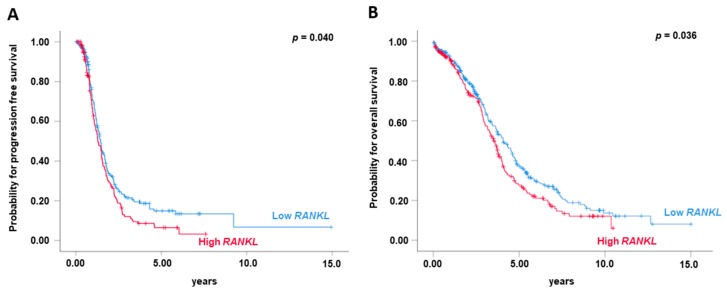
Validation of the prognostic impact of *RANKL* in the TCGA dataset. *RANKL* mRNA expression (*n* = 563) and (**A**) progression free survival and (**B**) overall survival.

**Figure 4 cancers-11-00791-f004:**
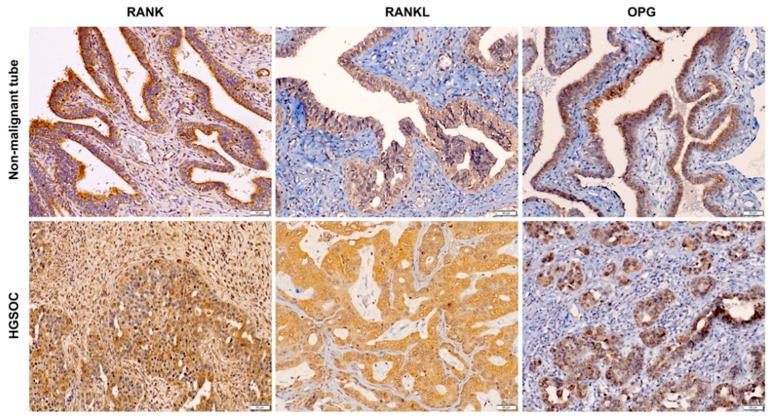
RANK, RANKL and OPG localize to cancer cells and tumour microenvironment in OC. Representative RANK, RANKL and OPG immunohistochemistry on FFPE sections from non-malignant tubes and HGSOC. *n* = 8–10 (per target); Scale bars indicate 50 µm.

**Figure 5 cancers-11-00791-f005:**
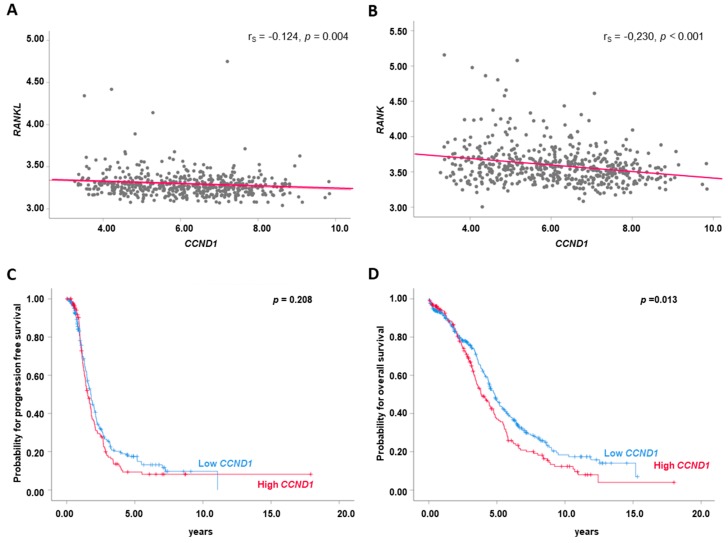
*RANK/RANKL* correlate with *CCND1* which is associated with clinical outcome in the TCGA cohort. Linear regression analysis of (**A**) *RANKL* and *CCND1*, (**B**) *RANK* and *CCND1* in tumour tissue of OC patients (*n* = 563). *CCND1* (*n* = 563) and (**C**) progression free survival and (**D**) overall survival.

**Table 1 cancers-11-00791-t001:** Univariate and multivariate survival analysis in 192 ovarian cancer patients. *RANK/RANKL/OPG* cut-off was determined by the Youden Index.

**Univariate Survival Analysis in 192 Ovarian Cancer Patients**							
**Variable**		**Progression Free Survival**			**Overall Survival**		
		**No. Patients**	**Median, Years**	***p* Value**	**No. Patients**	**Median, Years**	***p* Value**
		**(Relapsed/Total)**	**(95% CI)**		**(Died/Total)**	**(95% CI)**	
Age (median)	≤50.0 yrs.	21/32	2.13 (4.03–9.17)	0.762	15/32	9.15 (9.49–17.69)	0.019
	>50.0 yrs.	92/160	2.00 (6.71–10.11)		105/160	3.9 (7.00–10.04)	
FIGO stage	I/II	10/50	n.r. (13.63–18.83)	<0.001	18/50	n.r. (12.23–18.08)	<0.001
	III/IV	103/142	1.47 (4.03–6.95)		102/142	3.80 (6.07–9.26)	
Tumour grade	1–2	54/99	2.06 (6.98–11.34)	0.249	57/99	6.76 (9.09–13.78)	0.07
	3	57/91	1.98 (5.14–9.00)		61/91	3.71 (5.57–8.92)	
Residual disease after surgery	no	38/96	n.r. (10.54–15.10)	<0.001	35/96	n.r. (11.98–16.49)	<0.001
	yes	71/90	1.25 (2.25–5.02)		81/90	2.55 (3.74–6.62)	
Histology	HGSOC	81/122	1.81 (4.28–7.41)	0.003	88/122	3.70 (6.04–9.57)	0.001
	others	30/66	5.98 (9.31–14.68)		29/66	11.06 (10.31–15.68)	
*RANKL* mRNA expression	low	58/115	3.56 (1.81–5.31)	0.01	62/115	8.76 (5.98–11.55)	0.005
	high	55/77	1.68 (1.21–2.16)		58/77	3.62 (2.51–4.74)	
*RANK* mRNA expression	low	24/48	3.56 (0.89–6.24)	0.117	26/48	9.27 (1.97–16.56)	0.03
	high	89/144	1.97 (1.54–2.40		94/144	3.71 (1.85–5.57)	
*OPG* mRNA expression	low	101/177	2.06 (1.33–2.80)	0.135	107/177	5.80 (3.94–7.66)	0.154
	high	12/15	1.46 (1.16–1.76)		13/15	3.29 (1.87–4.70)	
Note: The significance level (*p*) was determined by log-rank test. *RANK/RANKL/OPG* cut-off was determined by the Youden Index. Abbreviations: CI, confidence interval; HGSOC, high grade serous ovarian cancer; n.r., not reached.							
**Multivariate Survival Analysis in 192 Ovarian Cancer Patients**							
**Variable**		**Progression Free Survival**			**Overall Survival**		
		**RR**	**(95 CI)**	***p* Value**	**RR**	**(95 CI)**	***p* Value**
Age	≤50.0 yrs >	1.19	(0.73–1.96)	0.489	1.54	(0.87–2.75)	0.134
FIGO stage	I/II vs. III/IV	3.57	(1.76–7.27)	<0.001	1.05	(0.58–1.90)	0.872
Tumour grade	1/2 vs. 3	1.09	(0.73–1.62)	0.682	1.08	(0.73–1.58)	0.705
Residual disease after surgery	no vs. Yes	2.01	(1.28–3.17)	0.003	4.96	(1.78–4.62)	<0.001
Histology	HGSOC vs. others	0.87	(0.55–1.36)	0.533	0.72	(0.46–1.12)	0.14
*RANKL* mRNA expression	low vs. high	1.42	(0.97–2.10)	0.017	1.7	(1.16–2.50)	0.007
*RANK* mRNA expression	low vs. high	1.19	(0.72–1.96)	0.502	1.36	(0.85–2.19)	0.202
*OPG* mRNA expression	low vs. high	1.11	(0.59–2.07)	0.749	1.25	(0.68–2.29)	0.466
Note: The significance level (*p*) was determined by Cox regression. *RANK/RANKL/OPG* cut-off was determined by the Youden Index. Abbreviations: CI, confidence interval; HGSOC, high grade serous ovarian cancer; RR, relative risk.							

**Table 2 cancers-11-00791-t002:** Multivariate survival analysis in 514 ovarian cancer patients from the TCGA Affymetrix dataset.

Variable		Progression Free Survival			Overall Survival		
		RR	(95 CI)	*p* Value	RR	(95 CI)	*p* Value
Age	≤ 50.0 yrs >	0.93	(0.83–1.05)	0.252	1.02	(0.95–1.11)	0.541
FIGO stage	I/II vs. III/IV	2.02	(1.38–2.97)	<0.001	1.14	(0.10–1.31)	0.060
Tumour grade	1/2 vs. 3	0.98	(0.89–1.09)	0.755	1.02	(0.94–1.11)	0.621
Residual disease after surgery	no vs. Yes	1.07	(0.96–1.20)	0.204	1.02	(0.92–1.14)	0.710
*RANKL* mRNA expression	low vs. high	1.32	(1.04–1.68)	0.022	1.26	(1.00–1.57)	0.046
Note: The significance level (*p*) was determined by Cox regression. Abbreviations: CI, confidence interval; RR, relative risk.							

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
