# Peer review of "Clinical Impact of RANK Signalling in Ovarian Cancer"

_cancers, 2019, doi:10.3390/cancers11060791_

Round 1
Reviewer 1 Report
Wieser et al., address the question of whether or not RANK/RANKL/OPG signaling plays a role in ovarian cancer. This work is important due to recent reports that RANK/RANKL/OPG signaling has effects on hormone-driven breast cancer, which means it may be important for ovarian cancer, which has similar genetics and is affected by hormones. They report an association with mRNA expression levels and PFS/OS, but, unfortunately, their functional assays did not identify a mechanism based on in vitrogrowth or response to chemotherapy.
Wieser, et al., report that RANKL/OPG mRNA expression is elevated in OC compared to healthy FT and a very slight increase in expression was also seen when comparing BRCA1/2 mutant tumors to BRCA1/2 w/t tumors (but not RANK). Furthermore, they report that dichotomization of OC cases by expression of RANK (<> 25thpercentile) or by expression of RANKL (<> 62ndpercentile) is an independent prognostic variable for survival, with high expression of either RANK/RANKL correlated with worse prognosis (roughly 10% difference in their discovery dataset and a much smaller difference in TCGA validation dataset). No correlation was seen with OPG. To confirm at the protein level, Wieser, et al., performed IHC on OC and compared to healthy FT. They did detect protein in all tissues, but did not detect a correlation with PFS/OS using semiquantitative analysis
Weiser et al., use Denosumab (anti-RANKL), cytokine stimulation, and addition of recombinant RANKL to test effect on cell proliferation using four OC cancer cell lines. No effect was seen on growth or response to chemotherapy.
Major Criticisms
Authors use Youden's J index to dichotomize samples. I am not a statistician, but it is unclear to me if this method is appropriate for a variable statistic like PFS and OS, as opposed to its use in determining an ideal ROC cut off for false positives/negatives associated with a question like, has/does not have cancer. As the authors indicate, the index returns quite different cutoff percentiles for the three genes. Authors calculate their cut-offs using Youden's index using their sample set and then apply it to the TCGA set as validation. To strengthen their hypothesis that RANK/RANKL/OPG signaling is clinically important, they should do the reverse analysis and calculate the Youden's J index using the TCGA set and then apply this to their dataset and report on the Kaplan Meier comparisons and significance.
Authors report no association of RANK/RANKL/OPG protein levels when comparing 20 OC tissues to non-malignant FT using IHC, but do not show the data. They also report a difference in cell-type expression comparing RANKL and RANK between stroma and epithelium, but do not show the data. This data is an important finding and should be reported in a supplementary table indicating IHC scoring of the three proteins on the various cell types in the sections analyzed. Because their functional data is negative, this IHC data will be most helpful for other researchers to determine how RANK/RANKL/OPG signaling might be affecting OC. A few more supplementary images highlighting these differences would also be helpful.
mRNA expression of RANKL/RANK/OPG was highly variable (Fig 3A-C & Supp) in the OC samples. Their IHC data indicated variable expression in stroma compared to epithelial cells. It is possible that the correlation of high expression of RANK/RANKL to OS/PFS is actually a reflection of the high levels being indicative of tumor purity in their samples, and the RANK/RANKL/OPG differences in mRNA only reflect differences in tumor purity. TCGA reports on tumor purity of samples based on IHC. Authors should calculate tumor purity in their 192 samples, or at least the 20 that they performed IHC on, and test for correlation between RANK/RANKL/OPG mRNA expression and tumor purity. Hopefully there would not be a strong correlation, and they could rule this out as a confounding variable.
Minor criticisms
No discussion or study of the two major downstream pathways of RANK/RANKL signaling are made (NFKb-CCND1, ID2-p21). The authors have access to RNAseq data for TCGA and could provide some analysis of the downstream genes (eg. CCND1 and p21) comparing their high to low RANKL OC's to get an idea of what pathway may be mediating the poor clinical outcomes.
Author Response
Please find our point-by-point response to the comments of reviewer 1 in the word file uploaded below.

Reviewer 2 Report
It would be nice to have some immunohistochemical data on normal ovarias and not only normal fallopian tubes
Table IA: “Median, months” should be “median, years”
The authors should expand the discussion on the immununomodulatory effects ofRANKL inhibitions. In the D-beyond study no effect on proliferation was seen in breast cancer but a significant increase in TILs. The potential role of immunotherapy (with or without densoumab) for the treatment of ovarian cancer should be discussed
The authors should consider to include the following references in the discussion referring to the potential use of RANKL inhibition (with or without immunotherapy) in vivo:
Nguyen B, Maetens M, Salgado R, venet D, Vuylsteke P, Polastro L, Wildiers H, Simon P, Lindeman G, Larsimont D, Van de Eynden G, Velgh C, Rothe F, Garaud S, Michiels S, Willard Gallo K, Azi HA, Loi S, Piccart M, Sotoriou C. D-BEYOND: a window of opportunity trial evaluating denosumab, a RANK ligand inhibitor and its biological effects in young pre-menopausal women diagnosed with early breast cancer. Cancer Res 2018;78(13 Suppl):abstarct nr CT101
Ahern E, Harjunpää H, Barkauskas D, Allen S, Takeda K, Yagita H, Wyld D, Dougall WC, Teng MWL, Smyth MJ. Co-administration of RANKL and CTLA4 Antibodies Enhances Lymphocyte-Mediated Antitumor Immunity in Mice. Clin Cancer Res. 2017 Oct 1;23(19):5789-5801. doi: 10.1158/1078-0432.CCR-17-0606. Epub 2017 Jun 20
Afzal MZ, Shirai K. Immune checkpoint inhibitor (anti-CTLA-4, anti-PD-1) therapy alone versus immune checkpoint inhibitor (anti-CTLA-4, anti-PD-1) therapy in combination with anti-RANKL denosumuab in malignant melanoma: a retrospective analysis at a tertiary care center. Melanoma Res 2018 doi: 10.1097/CMR.0000000000000459. [Epub ahead of print]
van Dam P , Verhoeven Y, Jacobs J, Wouters A, Tjalma W, Lardon F, Van den Wyngaer T, Dewulf J,Smits E , Colpaert C, Prenen H, Peeters M, Lammens M, Trinh XB. RANK-RANKL signaling in cancer of the uterine cervix. Int. J. Mol. Sci. 2019,20, 2183
Ahern E, Harjunpää H, Barkauskas D, Allen S, Takeda K, Yagita H, Wyld D, Dougall WC, Teng MWL, Smyth MJ. Co-administration of RANKL and CTLA4 Antibodies Enhances Lymphocyte-Mediated Antitumor Immunity in Mice. Clin Cancer Res. 2017 Oct 1;23(19):5789-5801. doi: 10.1158/1078-0432.CCR-17-0606. Epub 2017 Jun 20.
Smyth MJ, Yagita H, McArthur GA. Combination of anti-CTL-4 and anti-RANKL in metastatic melanoma. J Clin Oncol. 2016 Apr 20;34(12):e104-6. doi: 10.1200/JCO.2013.51.3572.
van Dam P, Verhoeven Y, Trinh XB, Wouters A, Lardon, F, Prenen H, Smits E, Baldewijns M, Lammens M. RANK/RANKL signaling inhibition may improve the effectiveness of checkpoint blockade in cancer treatment. Crit Rev Oncol Hematol 2019;133::85-91
Author Response
Please find our point-by-point response to the comments of reviewer 2 in the word file uploaded below.

Reviewer 3 Report
This study investigated RANK, RANKL and OPG expressions in 192 ovarian cancer tissues and 14 non-malignant control tissues, as well as ovarian cancer cell lines. The authors found that RANKL and OPG were highly expressed in ovarian cancer issue, whereas RANK expression was not elevated in OC compared to control. Impact of their expressions on ovarian cancer was also investigated using cohort of 192 ovarian cancer patients and an independent TCGA cohort. In a multivariate analysis, high RANKL expression was identified as an independent poor prognostic factor for PFS and OS. However, neither recombinant RANK ligation nor denosumab treatment affected ovarian cancer cell proliferation.
From the study results, the authors suggested that RANK signaling plays a significant role in the pathogenesis of ovarian cancer. The authors opened the possibilities of new drugs targeting RNAK signaling for the treatment of ovarian cancer.
The authors are commended for their comprehensive work. The topic is very interesting and the manuscript is well-written. Thank you for submitting the article for review. Please see the comments and questions below. Please submit your response per journal guidelines
1. Abstract. The term, “RANK constituents”, should be substituted to more specific terms.
2. Introduction should be re-written. Especially, Lines 56-63 is inadequate; they are results of the study. The aim of the study should be described in more detail.
3. The authors also used gene expression datasets from TCGA ovarian cancer. However, you should consider only serous ovarian carcinomas are included in TCGA.
4. You compared non-neoplastic fallopian tubes (control group) and ovarian tissue samples from ovarian cancer (Study group). More appropriate specimens in control group would be normal ovary.
5. What was the specific indications for surgery in control group?
6. For BRCA1/2 mutation status, did you consider only germline mutations? Or did you also include somatic mutations, too?
7. BRCA1/2 mutation status should be adjusted in multivariate analysis in ovarian cancer patients.
8. Lines 67. Non-neoplastic fallopian tubes of healthy controls vs. Lines 221. Elective salpingectomy for benign conditions. You should use the same or at least consistent term throughout the manuscript.
9. In the study population, there were various histologic types. Can you provide the study results confined to HGSOC? For example, in Results, ovarian cancer patients exhibited increased RANKL expression. Was it same among the patients with HGSOC?
10. It is interesting that RANKL mRNA expressions are elevated in ovarian cancer tissue, particularly in BRCA1/2 mutated ovarian cancer. Did you also investigate BRCA1 mutated ovarian cancer and BRCA2 mutated ovarian cancer, separately?
11. Lines 91-92. I understand you used the Youden Index to dichotomize the ovarian cancer cohort into high and low RANK, RANKL and OPG expressing tumors. However, please provide each cut-off value.
12. Lines 98-99. That result is very interesting. Can you provide the specific data?
13. Underlying mechanisms between RNAKL expression and poor clinical outcome should be further revealed. Inflammatory cytokines IL-1β and TNFα are not enough. Please suggest your hypothesis in detail. As the authors mentioned in Discussion, I think RANK signaling is also associated with BRCA-driven tumorigenesis in ovarian cancer, like breast cancer. That point could be one of the keys.
14. Typo. Line 48. Brca1 à BRCA1
Author Response
Please find our point-by-point response to the comments of reviewer 3 in the word file uploaded below.

Round 2
Reviewer 1 Report
Authors did a thorough and satisfactory job of responding to my comments. I have no further concerns
Reviewer 3 Report
Revised manuscript was much improved.
This study is acceptable.